# Design and development of sustainable and automated irrigation system based on multi-depth sensor fusion mechanism

Srikar Emany[+], Adithya P Rajeev 2[*], Anjali CV 3[*], and Sreehari V 4[*]

[+]IIT Palakkad, [*]IIT Palakkad

30 October 2023

**Abstract**

This project addresses the critical issue of irrigation water conservation by leveraging smart technologies and innovative methodologies to improve the accuracy and efficiency of irrigation systems. The existing irrigation practices often suffer from inaccurate measurements and the lack of automation, leading to inefficient water usage and potential environmental concerns. In response to these challenges, we introduce a standalone, fully automated irrigation model that requires no human intervention, thus minimizing the risk of human error. The proposed solution incorporates two key components to enhance irrigation efficiency: Multi-depth Soil Moisture Sensing and Spatial Interpolation to predict moisture distribution. The reliability and effectiveness of this approach have been well-established, and we are eager to implement it in practical settings to promote sustainable and efficient water usage in agriculture. This project represents a significant step towards addressing water scarcity and promoting responsible irrigation practices, benefiting both the environment and agricultural productivity. The sensor prototype was developed and tested in-house and the water conserved through this approach is estimated for inference. This proven method is ongoing vigorous testing at SATCARD (Smart Agri Tech Centre for Advanced Research and Development), IIT Palakkad.

## 1   Introduction

Irrigation is a vital component of India's agricultural landscape, serving as a linchpin for ensuring food security and economic prosperity. With over 50% of its workforce engaged in agriculture, the sustainable and eco-friendly management of this vital resource is of paramount importance. India's rich agricultural heritage is deeply intertwined with the need for precise and efficient irrigation practices, both to meet the ever-increasing demand for food and to alleviate the environmental pressures posed by water scarcity and over-exploitation.

Our focus is not merely to maintain the status quo but to bring about a transformation that empowers our farmers with the knowledge, tools, and technologies they need to adapt to the 21st century. A pivotal aspect of this transformation is to harness the potential of sustainable and eco-friendly agriculture. Achieving this goal entails not only preserving our environment but also enhancing the economic sustainability of our farmers.

Farmers are at the heart of India's agricultural sector, and empowering them is non-negotiable. They need the right information, support, and skills to navigate the ever-evolving world of technology and modern agriculture. It is imperative to provide them with access to the latest trends in agricultural technology, including smart irrigation solutions, to make their practices more cost-effective and efficient. We recognize that these hardworking individuals face numerous challenges, ranging from water scarcity to labor-intensive practices and the need to reduce their carbon footprint.

Moreover, it is essential that the technology intended to assist farmers is not only innovative but also accessible. It should be available at an affordable price point, ensuring that even the smallest-scale farmers can benefit from its potential.

In light of these pressing issues, we present a solution designed to revolutionize irrigation practices in India. Our proposal is centered around the automation of irrigation using a drip irrigation system equipped with a state-of-the-art multi-depth sensor fusion mechanism. This innovative approach relies on soil moisture sensors placed at different depths to accurately measure soil moisture data. To predict the moisture level of the entire agricultural plot, multiple sensors are strategically positioned and their data is analyzed using spatial interpolation techniques within QGIS software. This novel system has been implemented successfully and rigorously tested at IIT Palakkad's Smart AgriTech Centre for Advanced Research and Development (SATCARD).

The heart of this module is a cost-effective and reliable control system that autonomously manages the irrigation process. By controlling the pump and valves, this system reduces labor costs, optimizes water usage, minimizes wastage, and mitigates the risk of runoff, thus contributing significantly to both the prosperity of the farmers and the sustainability of our environment.

The novelty of this project lies in the integration of a multi-depth sensor fusion mechanism with the spatial interpolation technique to get the soil moisture distribution which will aid in facilitating automated irrigation systems. Studies have already been done on multi-depth sensors but studies that integrate it with an irrigation system by doing the spatial interpolation of soil moisture data are currently lacking.

## 2   Goals

- Efficient Water Discharge: To develop a state-of-the-art irrigation system that optimizes water usage by precisely controlling the volume of water discharged to agricultural fields, thereby minimizing water wastage and improving resource efficiency.

- Avoid Plant Water Stress: To implement an irrigation system that employs advanced sensors and algorithms to monitor soil moisture levels and plant water needs in real-time, ensuring that crops receive the appropriate amount of water to prevent water stress and enhance crop health and yield.

- Attaining Accurate Measurements: To achieve the highest level of measurement accuracy in soil moisture, weather conditions, and other relevant environmental variables, enabling the irrigation system to make data-driven decisions and adjustments to deliver water precisely where and when it is needed.

- Automated Irrigation: To create a fully automated irrigation system that operates without the need for manual intervention, reducing the labor and time required for irrigation management while enhancing consistency and effectiveness.

- Standalone Type Controller: To design and implement a standalone controller for the irrigation system, capable of operating independently and making autonomous decisions based on the collected data and predefined parameters, ensuring reliable and uninterrupted irrigation even in the absence of continuous human supervision.

## 3   System Architecture and Design

### 3.1   Hardware

Multidepth Sensor Fusion: The multi-depth sensor fusion mechanism is an innovative approach in agricultural technology, offering dual-depth measurement capabilities to assess soil moisture content. This advanced system facilitates more precise and data-driven irrigation practices by capturing critical information at varying depths within the soi, i.e., both primary and secondary sensing is performed. The data obtained from this technology serves two primary functions, each tailored to address the unique water requirements of different crops and soil conditions.

Firstly, the system leverages a dual-sensor approach to provide in-depth soil moisture measurement, enabling a high level of accuracy in predicting the water needs of specific plants. By simultaneously collecting

data from two sensors located at different soil depths, this two-point calibration method generates highly precise soil moisture measurements. This level of granularity allows farmers to make informed irrigation decisions, applying the right amount of water precisely where it is needed, thus minimizing both water wastage and the risk of overhydration.

The depth of crop root zone varies for different crops. Also, there are primary as well as secondary root zones. The consideration of both the root zones is important for providing irrigation. Hence, the incorporation of soil moisture sensors at multiple depths will help in estimating the soil moisture level more accurately within the range of primary and secondary root zones. In this study, two soil moisture sensors at multiple depths (15 cm and 30 cm) were used for the motive of measuring the soil moisture level accurately between the primary and secondary root zones. This will aid in the enhancement of crop water use efficiency. This adaptive approach ensures that the irrigation system accommodates the specific requirements of various plant types, promoting optimal growth and resource utilization. This mechanism stands as a testament to the versatility and efficiency that modern agricultural technology can bring to the field. By providing accurate, adaptable, and fine-grained soil moisture data, it paves the way for more sustainable and resource-conscious farming practices, while also enhancing crop yield and agricultural productivity. The moisture data from this mechanism is fed to QGIS software for spatial interpolation. Fig. 1 shows the multi-depth sensors installed at the SATCARD field.

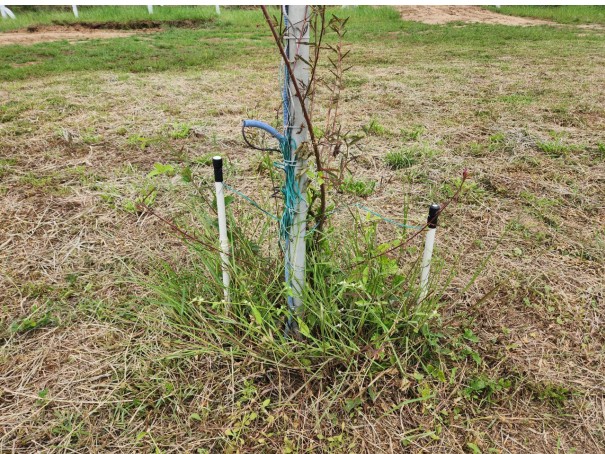

Figure 1: Soil moisture sensors at a single location at two different depths (15cm and 30cm).

Spatial Interpolation: Spatial interpolation plays a pivotal role in optimizing water management within the agricultural landscape, and its implementation at the SATCARD at IIT Palakkad serves as a prime example of modern precision agriculture. Within a 15-cent square plot, strategically positioned multi-depth sensor fusion soil moisture sensors are deployed at each corner and the central point of the area. These sensors continuously monitor soil moisture levels at various depths, collecting essential data required for efficient irrigation practices. The collected data is then subjected to Inverse Distance Weighting (IDW) interpolation, which is executed within the powerful geographic information system (GIS) software, QGIS. This process entails the spatial distribution of moisture data points, enabling the creation of a comprehensive moisture map for the entire 15-cent plot. Through spatial interpolation, the software accurately predicts soil moisture levels across the entire area, granting a holistic understanding of the land's moisture content. Fig. 2 shows the multi-depth sensors installed at SATCARD field.

The resulting moisture data is of paramount importance, as it serves as the foundation for an in-house automated control system. Designed and built specifically for this application, this system takes the interpolated data and seamlessly integrates it into the irrigation management process. It ensures the precise and autonomous control of the drip irrigation system across the plot, effectively minimizing water wastage, optimizing resource utilization, and enhancing crop health. The seamless integration of spatial interpolation, multi-depth sensor technology, and in-house automation exemplifies a cutting-edge approach to irrigation management, improving both sustainability and agricultural productivity.

Automated Control System: The revolutionary automatic control system for drip irrigation marks a signif-

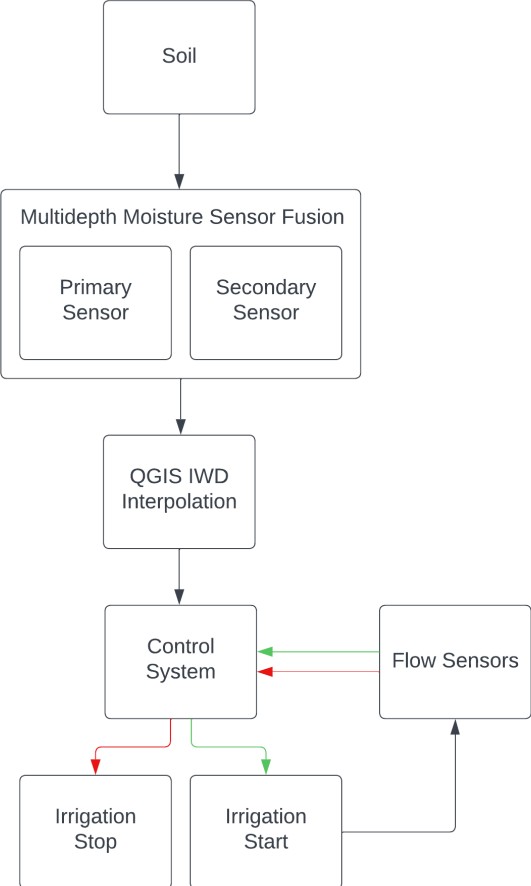

Figure 2: Block diagram representation of the operational flow of the System.

icant advancement in precision agriculture by seamlessly integrating data from multi-depth soil moisture sensors and QGIS interpolation techniques. This cutting-edge system guarantees the precise delivery of water to plants, reducing waste and optimizing resource utilization, all while eliminating the need for direct human intervention. At the core of this automated irrigation system lies the ESP32-based controller, serving as the central intelligence unit. With its robust computational capabilities and wireless connectivity, the ESP32 gathers and processes data from multi-depth soil moisture sensors distributed across the agricultural plot. These sensors provide real-time updates on soil moisture levels, ensuring that the irrigation system has access to accurate and up-to-date information. This system utilizes QGIS interpolation data to divide the plot into four distinct segments, each operated based on its unique moisture deficit input derived from the interpolation results. Tailored and automated irrigation schedules are crafted for each plot, finely tuned to meet the specific needs of the plants or crops in those areas. Using solenoidal valves, the ESP32 precisely regulates the flow of water into the drip irrigation system.

A pivotal component in this setup is the flow sensor, which continuously monitors the rate of water flow. It acts on feedback from the flow sensor, ensuring that the irrigation process stops when the desired flow rate is achieved. This proactive approach prevents over-irrigation, conserving water resources and enhancing efficiency. This comprehensive control system not only minimizes water wastage but also eliminates the need for manual irrigation management. By harnessing data-driven precision and the power of automation, it enhances both the efficiency of irrigation and the overall health and yield of crops. This makes it a cornerstone of sustainable and modern agricultural practices, delivering optimal results for each plot based on QGIS interpolation data while ensuring water conservation through flow sensor feedback.

| Component | Use |
| --- | --- |
| Soil Moisture Sensor | Primary Sensor at 15cm depth |
| Soil Moisture Sensor | Secondary Sensor at 30cm depth |
| Controller | ESP32-based control system with provisions for data logging (SD Card Module), time-based Pump On/Off feature (RTC Module), GSM Module for sending onsite operation information, and Solenoid Valve control circuitry |
| Flow Sensor | For reading water consumption for plots to send feedback to the controller for automatic valve cut-off when the required flow is reached |
| Solenoid Valve | DC Latching type valve (trigger-based) |

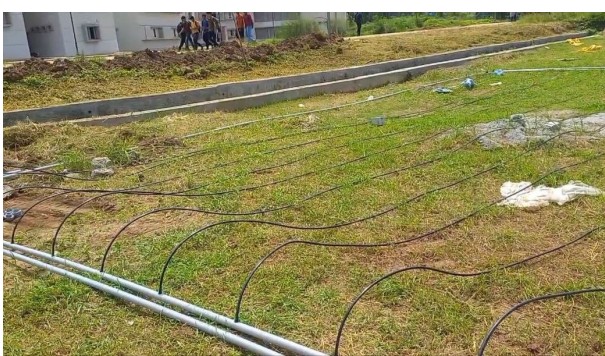

Figure 3: Drip Irrigation System at the field

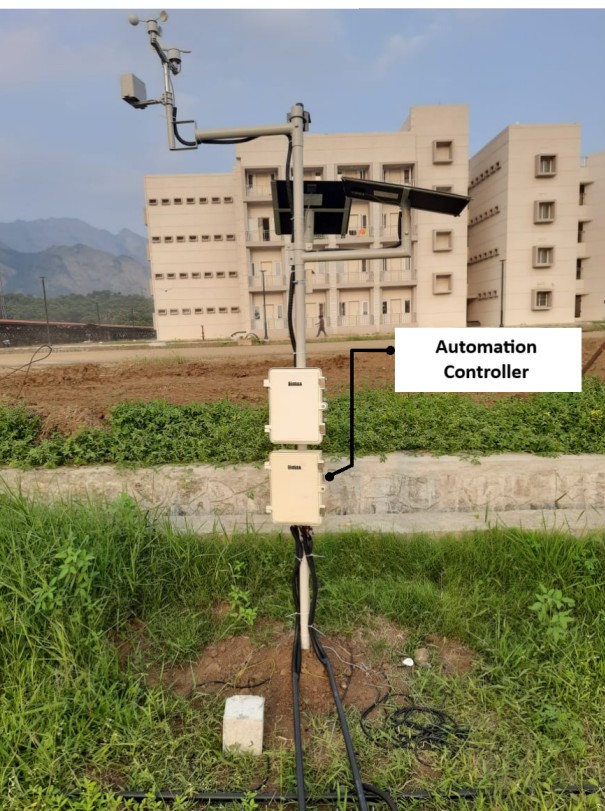

Figure 4: Automatic Drip irrigation controller installed on a pole with Weather station System.

# 4   Addressing Challenges

- Efficient Water Use: Optimise irrigation, conserve water. Intelligent systems aid this.

- Income Variation: Skill and resource disparities lead to income differences among farmers.

- Skill Development: Enhance skills, knowledge transfer, and modern practices to boost income.

- Resource Access: Varied land sizes, limited resources. Intelligent systems level the playing field.

- Sustainability: Efficient water use and sustainable practices vital for long-term farming viability.

# 5   Performance Evaluation and Testing Results

The evaluation of interpolated data was done practically using additional soil sensors for proper calibration as well as for testing the reliability of software-obtained data. These sensors were installed at known coordinate nodes for comparing the real-time moisture with the moisture value obtained with interpolation.

The methodology and results highlight the significant advantages of this system, which leverages advanced technologies to enhance irrigation efficiency while conserving water resources.

- Water Savings of Approximately 80%: One of the most striking results of our testing is the substantial water savings achieved through the implementation of automatic drip irrigation. Compared to conventional irrigation techniques, our system demonstrates water savings of approximately 80%. This remarkable reduction in water usage not only contributes to sustainable agricultural practices but also has economic benefits for farmers. By delivering water precisely to where it is needed, we mitigate the wastefulness associated with traditional irrigation methods, thereby promoting resource conservation.

- Enhanced Accuracy with Two-Level Soil Sensing and Interpolation: Our evaluation highlights the significance of utilizing two-level soil sensing and interpolation techniques. By deploying multi-depth soil moisture sensors and leveraging QGIS interpolation, we attain a higher degree of accuracy in measuring soil moisture levels. Traditional methods often rely on single-depth sensors or manual assessments, which can result in imprecise data. In contrast, our system provides real-time, multi-layer soil moisture information, offering a comprehensive understanding of the soil's moisture status. This enhanced accuracy allows for precise irrigation adjustments and tailored scheduling, optimizing plant health and crop yield.

- Automatic Drip Irrigation Based on Sensing Data: The core of our system's success lies in its capability to automate the irrigation process based on data from soil sensing. By continuously monitoring and analyzing the moisture levels within the soil, our system can dynamically adjust irrigation schedules to meet the specific needs of each plant or crop. This real-time responsiveness ensures that water is delivered only when required, preventing both under- and over-irrigation. The flow sensor, working in harmony with the soil sensing data, accurately regulates water flow and effectively stops irrigation when the desired moisture level is achieved. This integrated approach not only minimizes manual intervention but also guarantees optimal plant growth and resource utilization.

Regarding economic analysis, a comparative study between the conventional irrigation system and the designed automation system is done for the 15-cent test plot on the crop tomato.

For automated irrigation, only the capital investment is higher. The farmer will have to pay around ₹1.2 lakh as an initial payment. This can be done as a Pay-per-use model which will allow the farmer to invest monthly based on their usage which will be feasible for farmers who are not able for the initial high investment. However, with this system, the monthly expense of the farmer is reduced from ₹3040 to ₹1000. Since the monthly expense is less, the farmer will receive a payback of the initial investment of about ₹24480 per month, and thus after the completion of the payback, the farmer will be making a profit of around ₹24480 when compared to conventional irrigation method where this amount is a lost/monthly additional expense.

In summary, the performance evaluation and testing results confirm the remarkable benefits of our automatic drip irrigation system. It demonstrates significant water savings, achieves superior accuracy through two-level soil sensing and interpolation, and offers automated irrigation based on real-time sensing data. These findings underscore the system's potential to revolutionize agricultural practices, making them more sustainable, efficient, and environmentally responsible.

| Characteristics - Conventional | Physical Value |
| --- | --- |
| Available plot | 15 Cents |
| Irrigation requirement per day for 1 tomato plant | 0.4 litre |
| Total no of plants in 15 cents | 1100 plants |
| Total water requirement per day | 440 litres |
| No: of hours of irrigation per day (watering hose) | 1.52 hours |
| Labor charge per day (hourly charge) | ₹1000 |
| Total labor charge | ₹1520 |
| Electricity charge per unit | ₹4 |
| Pump Units Consumed | 380Wh |
| Total electricity to charge to run the pump | ₹1520 |
| **Total expense for conventional irrigation** | **₹3040** |

| Characteristics - Automated | Physical Value |
| --- | --- |
| Available plot | 15 Cents |
| Irrigation requirement per day for 1 tomato plant | 0.4 litre |
| Total no of plants in 15 cents | 1100 plants |
| Total water requirement per day | 440 litres |
| No: of hours of irrigation per day (automated) | 1 hour |
| Labor charge per day (hourly charge) | ₹0 (no manpower) |
| Electricity charge per unit | ₹4 |
| Pump Units Consumed | 250Wh |
| Total electricity to charge to run the pump | ₹1000 |
| **Total expense for automated irrigation** | **₹1000** |

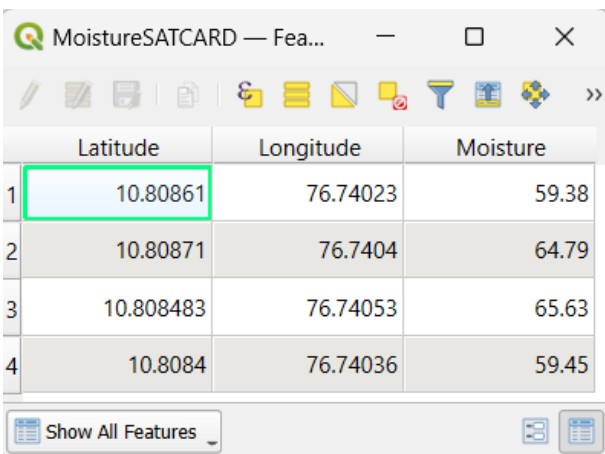

Figure 5: Coordinates of SATCARD field with moisture data obtained from soil moisture sensor.

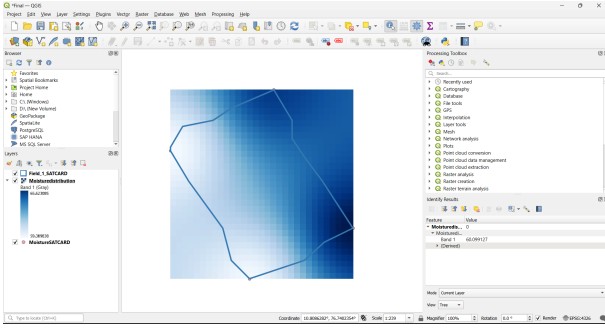

Figure 6: Spatial Interpolation was done with QGIS with the coordinates of the SATCARD test field. The average moisture data of each plot is obtained from the software and the value is sent to the irrigation controller.

## 6 Concluding Remarks and Avenues for Future Work

In our pursuit of advancing irrigation water conservation, the next crucial step in our future work is the implementation of wireless control for the irrigation system. By integrating wireless technology into our fully automated irrigation model, we aim to further enhance its efficiency and effectiveness. This wireless control system will enable remote monitoring and management of irrigation operations, allowing farmers to make real-time adjustments based on sensor data and environmental conditions.

Additionally, we plan to extend our research by exploring the integration of weather forecasting data and predictive analytics. By incorporating these elements, our irrigation model can proactively respond to impending weather changes, optimizing irrigation schedules to prevent over-watering or under-watering. This predictive capability will not only conserve water but also contribute to improved crop yields.

We intend to conduct field trials and gather feedback from local farmers to fine-tune our irrigation system and ensure its practicality and usability in real-world agricultural settings. Collaboration with industry partners and government agencies will be essential to scale up and implement our innovative irrigation solution across a wider agricultural landscape. Our ongoing and future work will continue to bridge the gap between technology and sustainable agriculture, offering a comprehensive solution to address water scarcity and promote responsible irrigation practices.

A GUI for the existing Weather stations at SATCARD is active and running. Here for weather predictions, weather data from different stations are being collected and processed. The drip irrigation system sensor data can be incorporated within this running algorithm as well as the GUI for enhancing the end-user experience as a future scope.

Since the study had been done for 15 cents, four sensors deployed on the four corners of the field were sufficient to carry out the analysis. In order to scale up the system, the number of sensors should be increased according to the area without the cost exceeding a certain limit. For that, an optimization needs to be done, which aims to find out the optimum number of sensors without exceeding the reasonable cost as well as meeting the demands of the plot.

## 7 Availability

https://drive.google.com/drive/folders/1N-YvC6hPNhpG2-BQHnu6HzOz6Z7WZrP7?usp=sharing

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
