# OpenReview forum: "Design and development of a sustainable and automated irrigation system based on a multi-depth sensor fusion mechanism"
_helsinki.fi/ESPC/2023/Competition — ESPC 2023 LongPresentation_

### Official Review · Reviewer_5xwx · 2023-11-14

**Rating:** 3
**Confidence:** 3

**Summary:**

Report presents the design and implementation of the automated irrigation system. System utilizes dual-sensor approach to measure the soil moisture at two depth levels. This data is then interpolated to the area of interest and fed to the irrigation management system.

**Strengths:**

1. Motivation is clear.
2. Design and implementation of the complete solution for automated irrigation management system.

**Weaknesses:**

Authors address important agricultural issue. There are few comments that could help in improving the report (or potential publications) afterwards.
1. Article is full of very strong claims which are not well supported by the analysis or references (examples "evolutionary automatic control system", "significant advancement in precision", etc)
2. I would suggest to provide picture highlighting the architecture of the system.
3. I'm not sure I fully understood how the interpolation was made, is it in 3D since there are 2 depths measured? Also, how is this data fed into the irrigation system if it's done in QGIS?
4. Are there technical details on how frequently the sensor data is acquired?
5. Performance evaluation and testing results - how the analysis is made? There is a claim that the system provides approx. 80% of water savings - how this was measured/what are the grounds for such conclusion?
Overall, interesting work that can be discussed at the event.

---

### Official Review · Reviewer_cnZB · 2023-11-14

**Rating:** 4
**Confidence:** 3

**Summary:**

The authors present a working prototype of a smart drip irrigation system that leverages multi depth sensors to monitor the moisture levels at various depths of the soil. Their prototype exemplifies how sensors can be leveraged for sensing the environment when taking decisions on how to use the resources efficiently.

**Strengths:**

- The team has a working prototype of a system that has been deployed in a field in the campus.
- The system takes into account the varying needs from the consumers. Specifically, the multi-depth sensors highlight that the system can be used in a wide variety of fields

**Weaknesses:**

- It is not clear how the authors arrived at the 80\% improvement in the water consumption. What was the base-line (conventional irrigation) against which the improvement was measured? Does conventional irrigation include existing drip irrigation solutions?
- The video could have been longer with some more details on the evaluation and the challenges faced when deploying the sensors.
- The authors have presented limited details on the control loop and decision process.  For instance, how did the authors determine the appropriate moisture levels that trigger the start and stop of irrigation?

Here are some suggestions that might be useful to further strengthen this work.
- Critical Analysis on the factors that affect the deployment of the sensors. For instance, when would having multiple sensors (beyond two) be useful?
- Challenges in calibrating the sensors and how they can be recalibrated. For instance, does the type of fertilizers or the chemicals used in pesticides affect the sensors and readings?

---

### Official Review · Reviewer_3oib · 2023-11-17

**Rating:** 2
**Confidence:** 4

**Summary:**

The project presents fully automated irrigation model. The proposed solution incorporates two key components: Soil monitoring and irrigation actuation. Soil moisture is done in two ways: multi-depth at few locations and spatial Interpolation to predict moisture distribution across the measurement region. The sensor prototype was developed and tested in-house and the water conserved through this approach is estimated for inference.

**Strengths:**

1. Block diagram is good.
2. Field installation and professional deployment of automated control system and soil measurement.

**Weaknesses:**

1. Presentation could had been bit better.
2. How many nodes are deployed? How was evaluation of interpolation conducted?
3. No results are shown except for one table of values.
4. Motivation for multi-depth not given. Also, it is not clear how it is used finally.

---

### Official Review · Reviewer_Zifg · 2023-11-18

**Rating:** 2
**Confidence:** 4

**Summary:**

The project presents the automation of irrigation using a drip irrigation system equipped with a state-of-the-art multi-depth sensor fusion mechanism. This automation system relies on soil moisture sensors placed at different depths to accurately measure soil moisture data. To predict the moisture level of the entire agricultural plot, the project deployed sensors in four positions and analyzed the sensor data using spatial interpolation techniques within QGIS software. The main goal of the project is to propose a solution for sustainable irrigation of agricultural plots.

**Strengths:**

The well-written and well-presented report. A practical system that uses multi-depth soil sensors, spatial interpolation, and an automated control system.

**Weaknesses:**

Despite the nice work, the novelty of this project has remained unclear. The project does not address the previous solutions, and then it does not compare the proposed solution against the existing ones.
In addition, it seems that in the current deployments, there are only four points where the sensors are installed. The project does not estimate the number of sensors to be deployed within a certain area.
The cost analysis of such sensor deployments is also missing. It is not clear if the proposed system offers an economically beneficial solution. The report also does not investigate the amount of water saving compared to the traditional way of soil irrigation.
Moreover, the report does not explain detailed information about the soil sensor and the controller board. The report does not explain the control system well enough. It also does not explain how they perform calibration using the data obtained from two sensors in two different depths (15cm and 30 cm).

---

### Official Review · Reviewer_xTVU · 2023-11-18

**Rating:** 4
**Confidence:** 3

**Summary:**

In this project, authors have attempted to address the inefficiencies in current irrigation practices.
The solution given is fully automated and helps farmers make informed decisions for better irrigation decisions.
Their system has two major components -- Multi-depth Sensor Fusion for soil moisture sensing and Spatial Interpolation to predict moisture distribution.
Multi-depth sensor fusion helps in predicting the water needs of specific plants as well as depth of their roots.
Spatial Interpolation sensors monitor soil moisture levels at different depths. This data is then exposed to IDW interpolation within a GIS software. As an end result, a comprehensive moisture map is prepared for the field, which serves as the basis of the automated irrigation control system.
With this system, they are able to get water savings of up to 80\% and enable an intelligent and automated irrigation system in Indian farms.

**Strengths:**

The report is well-presented and easy to read.
The developed system has a high societal impact by improving farming practices and saving crucial water.
It has addressed a pressing problem of Indian farming. When implemented at a large-scale, it will be phenomenal.

**Weaknesses:**

*A nice GUI for end users would be helpful
*Incorporating machine learning algorithms can help predict future needs as well as plant specific needs in correlation with environment factors such as, weather changes